# Improving Visual-Patient-Avatar Design Prior to Its Clinical Release: A Mixed Qualitative and Quantitative Study

**DOI:** 10.3390/diagnostics12020555

**Published:** 2022-02-21

**Authors:** Doreen J. Wetli, Lisa Bergauer, Christoph B. Nöthiger, Tadzio R. Roche, Donat R. Spahn, David W. Tscholl, Sadiq Said

**Affiliations:** Institute of Anaesthesiology, University Hospital Zurich, Raemistrasse 100, 8091 Zurich, Switzerland; doreenjessica.wetli@usb.ch (D.J.W.); lisa.bergauer@usz.ch (L.B.); christoph.noethiger@usz.ch (C.B.N.); tadzioraoul.roche@usz.ch (T.R.R.); donat.spahn@usz.ch (D.R.S.); sadiq.said@usz.ch (S.S.)

**Keywords:** avatar technology, intuitiveness, patient monitoring, situation awareness, user-centred design, Visual-Patient-avatar

## Abstract

Visual-Patient-avatar, an avatar-based visualisation of patient monitoring, is a newly developed technology aiming to promote situation awareness through user-centred design. Before the technology’s introduction into clinical practice, the initial design used to validate the concept had to undergo thorough examination and adjustments where necessary. This mixed qualitative and quantitative study, consisting of three different study parts, aimed to create a design with high user acceptance regarding perceived professionalism and potential for identification while maintaining its original functionality. The first qualitative part was based on structured interviews and explored anaesthesia personnel’s first impressions regarding the original design. Recurrent topics were identified using inductive coding, participants’ interpretations of the vital sign visualisations analysed and design modifications derived. The second study part consisted of a redesign process, in which the visualisations were adapted according to the results of the first part. In a third, quantitative study part, participants rated Likert scales about Visual-Patient-avatar’s appearance and interpreted displayed vital signs in a computer-based survey. The first, qualitative study part included 51 structured interviews. Twenty-eight of 51 (55%) participants mentioned the appearance of Visual-Patient-avatar. In 23 of 51 (45%) interviews, 26 statements about the general impression were identified with a balanced count of positive (14 of 26) and negative (12 of 26) comments. The analysis of vital sign visualisations showed deficits in several vital sign visualisations, especially central venous pressure. These findings were incorporated into part two, the redesign of Visual-Patient-avatar. In the subsequent quantitative analysis of study for part three, 20 of 30 (67%) new participants agreed that the avatar looks professional enough for medical use. Finally, the participants identified 73% (435 of 600 cases) of all vital sign visualisations intuitively correctly without prior instruction. This study succeeded in improving the original design with good user acceptance and a reasonable degree of intuitiveness of the new, revised design. Furthermore, the study identified aspects relevant for the release of Visual-Patient-avatar, such as the requirement for providing at least some training, despite the design’s intuitiveness. The results of this study will guide further research and improvement of the technology. The study provides a link between Visual-Patient-avatar as a scientific concept and as an actual product from a cognitive engineering point of view, and may serve as an example of methods to study the designs of technologies in similar contexts.

## 1. Introduction

The World Health Organisation (WHO) regards perioperative complications as a significant global cause of preventable death in healthcare [1]. Consequently, the guidelines for safe surgery describe continuous patient monitoring by anaesthesia providers as “extremely important” for patient safety during surgical procedures. However, healthcare providers must continuously perceive and interpret many monitoring values to gain and maintain situation awareness, and initiate possibly lifesaving therapies. The central concept of well-informed decisions is situation awareness, defined as perceiving and understanding the current situation, including its expected course [2,3,4]. Failure to perceive principally available information is more common than misinterpreting information or inferring incorrect projections from it [5]. Research showed that up to eighty percent of adverse events in anaesthesia result from impaired situation awareness [6].

Although these problems are well known and the design of patient monitoring holds opportunities to increase situational awareness [7], there have been no significant improvements in information presentation since the mid-20th century [8]. Current patient monitoring relies on numbers and waveforms, which humans can only read one-by-one, and can have difficulty memorising and integrating mentally [3,7,9]. The resulting delay in understanding clinical situations [7] calls for a new monitoring approach. 

### Visual-Patient-Avatar

The Visual-Patient-avatar is a visualisation technology for vital sign monitoring in patient monitors, as used in intensive care units or operating rooms. It was developed based on user-centred design [2,10] and cognitive engineering principles [11,12], in order to increase care providers’ situation awareness. It does so by improving the exchange of information at the human-machine interface. The technology creates a virtual model of the monitored patient from the monitoring data. The information content is exaggerated but presented realistically. When blood pressure is high, the avatar’s body begins to pulse firmly; when oxygen saturation is low, it turns purple, and when brain activity is low, it closes its eyes. This type of display eliminates the need for users to translate the monitor’s numbers before interpreting them cognitively, and is better adapted to the strengths of human sensory perception and information processing [13,14,15]. In several computer-based studies, Visual-Patient-avatar facilitated time-efficient monitoring, increased the users’ diagnostic confidence and lowered their perceived workload [13,14,15,16,17]. In addition, Visual-Patient-avatar improved the ability to monitor multiple patients simultaneously [18], and was described as intuitive and easy to learn [19,20,21]. A recent simulation study found quicker recognition of the cause of simulated emergencies when teams used avatar-based monitoring [22]. The previous studies were conducted with the initial prototype design created at the University of Zurich. Although this design was functionally good, design-wise, it was very rudimentary and not optimally proportioned. For example, the head was too large and the thorax too small. Since an aesthetically optimised appearance may improve care providers’ acceptance and a technology’s perceived usability [23], the initial design of Visual-Patient-avatar required adaptions in terms of functionality and aesthetics before its introduction to clinical reality. 

This study investigated first impressions of avatar-naive anaesthesia providers regarding the original design of Visual-Patient-avatar. Based on these findings, the avatar was redesigned. The study then explored general acceptance of the new, revised design and understanding of the redesigned visualisations of avatar vital signs in a new group of avatar-naive participants. This study aimed to improve the acceptance and functionality of Visual-Patient-avatar technology through an evidence-based redesign process.

## 2. Methods

The responsible ethics committee reviewed the study protocols and issued declarations of non-responsibility (Business Management System for Ethics Committees numbers Req-2020-00059 and Req-2021-00393). All participants provided written informed consent for the anonymous use of their data.

### 2.1. Study Design and Participants

This was an investigator-initiated, prospective, single-centre, mixed qualitative and quantitative study conducted at the University Hospital Zurich in Switzerland, a tertiary care facility. This study consisted of three separate parts; a flowchart is provided as Figure 1. The first, a qualitative study, aimed to explore anaesthesia providers’ first impressions regarding the original, unrevised design of Visual-Patient-avatar. Structured interviews were conducted with Visual-Patient-avatar-naive subjects. After a qualitative interview analysis, the second study part, the redesign followed, in which the avatar’s design was adapted based on the obtained results. The third, final part of the study aimed to quantify general acceptance of the new, revised design, and the level of intuitive recognition of the redesigned vital sign visualisations. The first and third study parts included participants who worked either as resident or staff anaesthesiologists, or as anaesthesia nurses in the designated study centre. Only participants without previous contact with Visual-Patient-avatar were included.

### 2.2. The First, Qualitative Study Part

#### 2.2.1. Description Qualitative Interview

For the qualitative first part of the study, four Visual-Patient-avatar scenarios reflecting different patient conditions before or during general anaesthesia were presented. In a quiet and distraction-free environment, participants were instructed to express aloud their feelings, thoughts and ideas while looking at the scenarios. Appendix A depicts all four scenarios in detail, Appendix A provides the exact wording of the standardised instructions and Appendix A provides an example video of a participant interview. The scenarios were presented in sequence for one minute each, on a Philips IntelliVue MX500 patient monitor. Participants verbalised their reflections in their native German language, so that no potential language barrier limited their thought processes. Participants were informed that no further information on Visual-Patient-avatar would be provided until after the study session, in order to prevent influencing results. Audio of the structured interviews was recorded using a Sony ICD-UX560 voice recorder (Sony Corp., Minato, Tokyo, Japan). Subsequently, all audio recordings were automatically transcribed using the transcription programme fx4 (Dr. Dresing & Pehl GmbH, Marburg, Germany). Following the transcription rules listed in Appendix A, the automatically created transcript was revised manually. To qualitatively analyse the transcripts, the template approach [24], a highly structured method for qualitative research, was applied. Major topics with hierarchical subthemes were identified based on word count using Microsoft Word (Microsoft Corporation, Redmond, WA, USA) and free inductive coding based on recurring participants’ answers. The resulting template was modified until the coders agreed upon a final version. Figure 2 shows this final coding template, which was then applied to investigate the obtained data. The decision to conduct a think-aloud-like study without training participants on the technology beforehand was based on the belief that unbiased, non-pre-primed thoughts are particularly valuable for identifying the strengths and shortcomings of a technology. This procedure creates a situation like when a person walks into an intensive care unit or operating room and sees the avatar on the screen for the first time. It was considered worthwhile to use the opportunity as long as it would still be possible. Once it is introduced in reality, videos will circulate on the Internet, and care providers will have pre-formed opinions or have learned how to interpret certain visualisations.

#### 2.2.2. Analysis of Vital Sign Visualisations

Further transcription analysis focused on the intuitiveness of vital sign visualisations. The transcripts were examined for unsolicited, spontaneous statements about the depicted vital signs. This analysis assessed whether interpretations were, in the intended sense, non-specific (without assuming a possible meaning) or incorrect. This assessment aimed to provide insight into which illustrations may benefit from a redesign for better intuitive understanding.

### 2.3. The Second, Redesign Study Part

Based on the identified improvement opportunities from the first, qualitative study part, the avatar was redesigned in collaboration with Philips’ design and technical specialists between June 2020 and April 2021. This process reviewed the overall appearance of Visual-Patient-avatar, including all vital sign visualisations.

### 2.4. The Third, Quantitative Study Part

#### 2.4.1. Description of Quantitative Assessment

In this computer-based study part, the newly revised design was quantitatively analysed with a new group of anaesthesia providers. An iSurvey-based questionnaire (Harvest Your Data, Wellington, New Zealand) running on an iPad (Apple Inc., Cupertino, CA, USA) was used to input data. Again, all data were collected in a quiet, distraction-free setting. In a standardised manner, outlined in Appendix A, the basic concept of Visual-Patient-avatar was explained without clarifying each vital sign in detail. The participants were asked to answer four questions based on themes identified in the previous first, qualitative study part, and during the redesign process. The first question asked about the preferred orientation of Visual-Patient-avatar on the monitor screen (answer options: upright, 45° clockwise rotation, 135° clockwise rotation with head down). The second question used a five-point Likert scale from disagree to agree to assess the level of consent with the statement that the avatar’s revised skin colour reflects all of humanity. The third question asked about the preferred size of Visual-Patient-avatar using a split-screen mode, where conventional and avatar monitoring modalities are displayed side-by-side on the screen (answer options: 50%, 25% or 12.5% of the screen area). The fourth question used a five-point Likert scale from disagree to agree to evaluate whether the avatar’s appearance looks professional enough for medical use. The answers are presented as proportions and percentages. For the skin colour and design professionalism questions, the Wilcoxon signed-rank test was used to determine whether the sample medians differed significantly from neutral. A *p*-value of <0.05 was considered to be statistically significant. Microsoft Excel and Prism 8.4.1 (GraphPad Software, San Diego, CA, USA) were used to display and analyse the quantitative data.

#### 2.4.2. Analysis of the Vital Sign Visualisations

The participants were also asked to identify twenty vital sign visualisations in a randomised order using videos with the revised design. Appendix A provides an example video. These videos were structured in such a way that the avatar was first shown in a normal vital sign state, and then with a single changed vital parameter. Using the iSurvey questionnaire, the participants indicated which vital sign visualisation they thought had changed. The results of these questions are presented as proportions and percentages. For the analysis, we used Microsoft Excel and Prism 8.4.1.

## 3. Results

### 3.1. Study and Participant Characteristics

Between 4 May and 28 May 2020, 51 interviews with 51 participants were conducted for the first, qualitative study part, all of which were transcribed and analysed. Between 12 April and 20 April 2021, 30 new participants were recruited for the third, quantitative study part. 99% (718 of 720) of the questions were answered, and all questionnaires (30 of 30) analysed. Table 1 provides an overview of the study characteristics.

### 3.2. The First, Qualitative Study Part

#### 3.2.1. Interview Analysis

The Consolidated Criteria for Reporting Qualitative Research (COREQ) [25] and the Standards for Reporting Qualitative Research (SRQR) [26] were followed in this study. Two main themes with two subthemes each were identified, and participant statements from the transcribed interviews were grouped accordingly. Figure 2 depicts the generated coding template. Table 2 illustrates the transcripts’ qualitative analysis using the coding template.

#### 3.2.2. Vital Sign Visualisation Analysis

In the interview transcripts, 756 statements mentioning vital signs were identified. Appendix A breaks down the comments by vital sign and scenario. Appendix A contains the complete dataset for the vital sign analysis. The most frequently correctly interpreted vital sign was body temperature, with 76% (38 of 51) accurate statements. All other vital signs were rated correctly in less than half of the statements. Central venous pressure was the vital sign with the fewest accurate statements (5 of 51 [10%] statements).

### 3.3. The Second, Redesign Study Part

Based on the results of the first, qualitative analysis, Visual-Patient-avatar was redesigned from July 2020 until the end of March 2021. Since the participants often criticised the avatar’s shape, we modified its body proportions to resemble the human shape more closely. Among others, the thorax area was enlarged to allow better visibility of the illustrations in the thoracic area. As participants also commented on the avatars’ original spatial orientation (135° clockwise rotated with the head down), two additional positions were created with an upright avatar and a 45° clockwise rotated position. Furthermore, several vital sign visualisations were redesigned to increase their intuitive recognisability. This was accomplished by improving the anatomical correctness of the representations. For example, the illustration of central venous pressure, which initially was frequently misrecognised, was designed more anatomical. Finally, the avatar’s skin colour had to be adapted from a Caucasian skin tone to a darker complexion to cover a larger part of global population. Figure 3 shows Visual-Patient-avatar before and after the redesign.

### 3.4. The Third, Quantitative Study Part

#### 3.4.1. Survey Analysis

The general questions had a 100% response rate. Concerning the spatial orientation, 28 of 30 (93%) participants preferred the upright or nearly upright orientation. When asked about the avatars’ skin colour, 16 of 30 (53%) participants rather agreed or agreed to the statement that the colour was a good choice. Using the Wilcoxon signed-rank test, these answers differed significantly from neutral, with *p* = 0.008. Regarding the split-screen monitor, 28 of 30 (93%) participants favoured a Visual-Patient-avatar size of 25% of the monitor or more. Twenty of 30 (67%) participants rather agreed or agreed to the statement that the redesign looks professional enough for medical use. The Wilcoxon signed-rank test showed that mentioned answers differed significantly from neutral, *p* < 0.001. Figure 4 shows a detailed overview of the quantitative survey results.

#### 3.4.2. Vital Signs Visualisation Analysis

Figure 5 shows a detailed analysis of intuitive interpretation accuracy concerning vital sign visualisations. In total, the participants rated 600 visualisations. Two of 600 (0.3%) answers were missing due to app-based technical issues and 435 of 600 (73%) answers were correct. Twenty-three of 30 (77%) participants misinterpreted the two vital sign visualisations, depicting low blood pressure and ST-segment deviations. The anaesthesia personnel intuitively recognised the other 18 vital sign changes with accuracies of 57% or higher. Central venous pressure, which had hardly been recognised in the first study part, performed well in the quantitative analysis after its redesign. Low central venous pressure was interpreted correctly in 26 of 30 (87%) answers. High central venous pressure was identified correctly in 27 of 30 (90%) answers.

## 4. Discussion

This mixed-method study’s main objective was to render the redesign of Visual-Patient-avatar a scientific process. Based on the qualitative results of the structured interviews, the avatar’s shape, spatial orientation, skin colour and several vital sign visualisations were redesigned. After the redesign process, a quantitative assessment found high approval regarding the modifications.

The interview analysis succeeded in revealing design aspects that participants recurrently identified as disturbing or unclear. Redesigning these issues was considered critical for ensuring the acceptance and usability of Visual-Patient-avatar. The original version depicted the avatar in a comic-style design with a spatial perspective that turned the avatar upside down to mirror the anaesthesiologist’s usual perspective at the patient’s head. As participants repeatedly commented on these aspects, two new versions with the avatar’s head pointed upwards were designed. The main design focus was on anatomically correct representations with simplified body parts while maintaining realistic proportions. The intention behind the simplified, cartoon-style presentation was to avoid more complex design elements that would potentially distract from essential information and produce a mental workload to interpret its meaning. A guiding principle of user-centred design is to narrow down presentation to the most critical information based on users’ objectives [2].

The share of Visual-Patient-avatar on the monitor is a relevant aspect for its clinical introduction. Seventy-three percent of participants preferred 25% of the monitor screen area. This proportion seems in line with other situation awareness-oriented visualisations in medical devices, such as the dynamic lung product available in Hamilton ventilators (Hamilton Medical AG, Bonaduz, Switzerland) [27,28].

One of the main design modifications concerned the avatar’s skin colour. As Visual-Patient-avatar technology is intended for global use, it became clear that a revision of skin tone was necessary. The quantitative survey showed that no one completely disagreed and 16 of 30 (53%) participants rather agreed or agreed the redesign was a good compromise for the representation of all humanity. Supported by this result, this new skin colour was kept for the clinical rollout of Visual-Patient-avatar. This modification is significant, considering global efforts to achieve equality for all ethnic groups. Selecting one colour for everyone seemed more desirable than introducing different colours and having to switch between them.

In the original design, some vital sign visualisations proved difficult to interpret without explanations. According to the quantitative assessment of the revised design, the more anatomically correct vital sign modifications led to a 73% probability to intuitively recognise the visualisations in their intended sense, where ‘intuitively’ means without any teaching or training whatsoever. Other researchers evaluating user-centred interfaces also observed that anatomically accurate representations promote intuitive understanding [10]. In medical technology, the question regarding the relevance of intuitive understanding often arises [29,30], as some training can usually be expected. In the past, however, research has attributed counterintuitive design as a factor responsible for the failure of medical visualisation technologies [7], which is why this aspect remains relevant. According to the definition of Blackler and colleagues, intuition is a type of cognitive processing that is often, but not always, non-conscious, and utilises stored experiential knowledge [31]. In previous studies, anaesthesia and intensive care personnel quickly understood the avatar visualisations after obtaining a short instructional video [14,15,16,18,19,22,32]. Even after the redesign, however, there remain ambiguities in the intuitive assessment of some vital signs, especially ST-segment deviations and low blood pressure. Therefore, despite its simple design features geared at intuitive understanding, some Visual-Patient-avatar teaching should be provided before clinical use. The level of intuitive understanding found in this study suggests that the technology succeeds in reaching the goal of being a tool designed to promote situation awareness, which is to transfer the relevant information as quickly as possible, with the lowest possible cognitive stress, while easing adoption and use [2]. The clinical importance of increasing situation awareness is evident in the relationship between situation awareness breakdowns and adverse events [3,4,5,6].

### Limitations and Strengths

This study had several limitations. First, this was a monocentric central European study conducted in the hospital where Visual-Patient-avatar was invented. Opinions may differ elsewhere in the world. Second, even though the study was conducted according to the COREQ [25] and SRQR [26] criteria, qualitative research inherently includes subjective judgment in data collection and analysis. As both the study team and the participants consisted of anaesthesia personnel, this shared background may have influenced the subjective assessments. Furthermore, the measured types of intuitive vital sign interpretation in the first, qualitative and third, quantitative part are not the same, and thus a direct comparison is not possible. In the first, qualitative study part, the participants were intentionally not instructed to interpret the vital signs, in order to avoid investigator-induced mental preconceptions; this meant that respondents did not always attempt to interpret each visualisation. Thus, this part of the study enabled us to get an impression of the participants’ unconscious intuitions. According to the definition of intuition of Blackler and colleagues [31], in the third, quantitative study part, conscious intuition was measured, as the participants were explicitly instructed to interpret the visualisations.

The qualitative interview approach is not just a limitation. It is one of the study’s strengths, as it allowed us to obtain unbiased and broad feedback, while the survey enabled us to weigh relevant aspects of the identified themes quantitatively. Another strength is the balanced sample in terms of gender and occupation in both study parts. These facts and the high response rate reduced a possible sample bias.

## 5. Conclusions

Using an innovative study approach allowed us to improve and fine-tune the design and concept of Visual-Patient-avatar. Intuitiveness and perceived professionalism achieved a good level in the new, more anatomically correct design. A head-up rotated version with a screen size of 25% seemed to be preferred by anaesthesia personnel, and may therefore serve as the default version. Future studies may further tweak the few visualisations that still had low intuitiveness after the redesign, and may analyse the effects of giving them special consideration in instructional material. Future studies must continue to evaluate the performance of the concept and design in real-life clinical use. These studies should incorporate feedback from a global population, and continue to refine vital sign illustrations with the goal of achieving an avatar product that maintains its functionality, and is accessible for the broadest possible group of professional healthcare providers and everyone who may come into contact with it, e.g., patients and their family members.

## Figures and Tables

**Figure 1 diagnostics-12-00555-f001:**
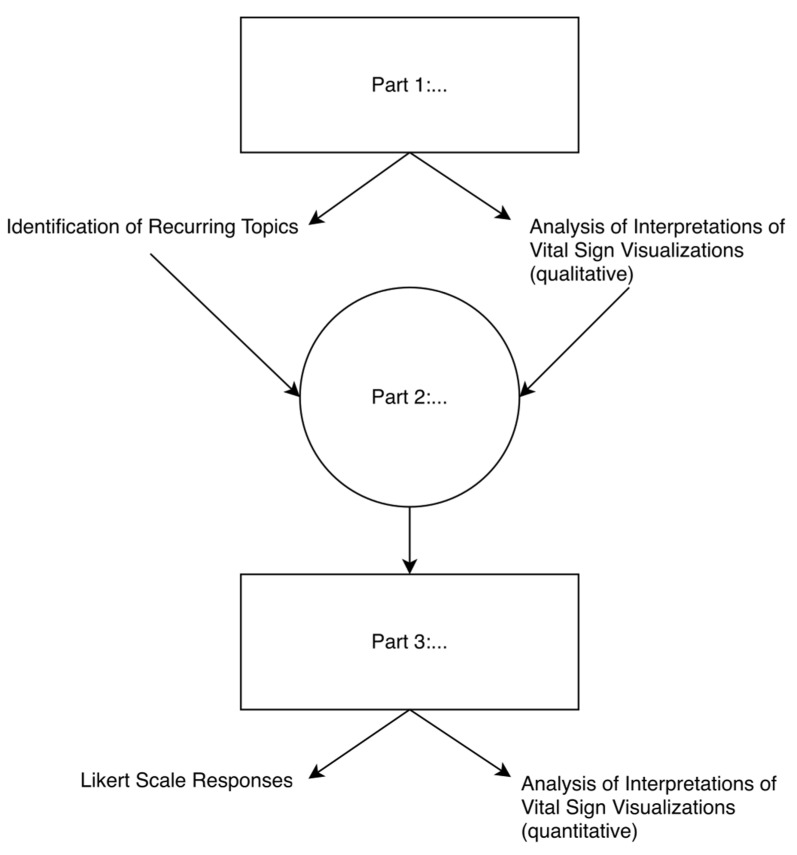
A flowchart outlining the study’s design, methodology and parts.

**Figure 2 diagnostics-12-00555-f002:**
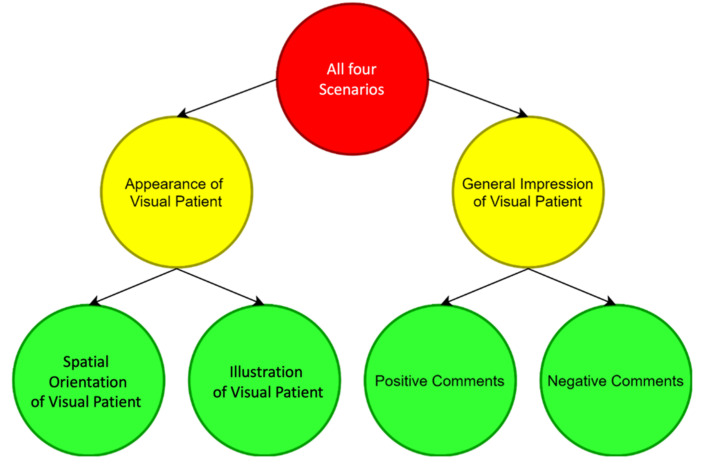
The coding template generated using deductive coding and free inductive coding based on recurring answers in the structured interview transcripts.

**Figure 3 diagnostics-12-00555-f003:**
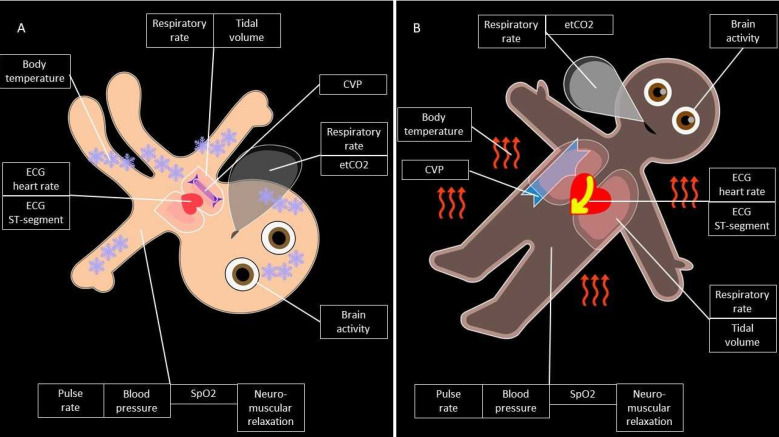
The Visual-Patient-avatar before (**A**) and after (**B**) the redesign process. CVP: central venous pressure, ECG: electrocardiogram, etCO2: end-tidal carbon dioxide, SpO2: peripheral oxygen saturation. When the body temperature is within the normal range, neither ice crystals nor heatwaves are visible.

**Figure 4 diagnostics-12-00555-f004:**
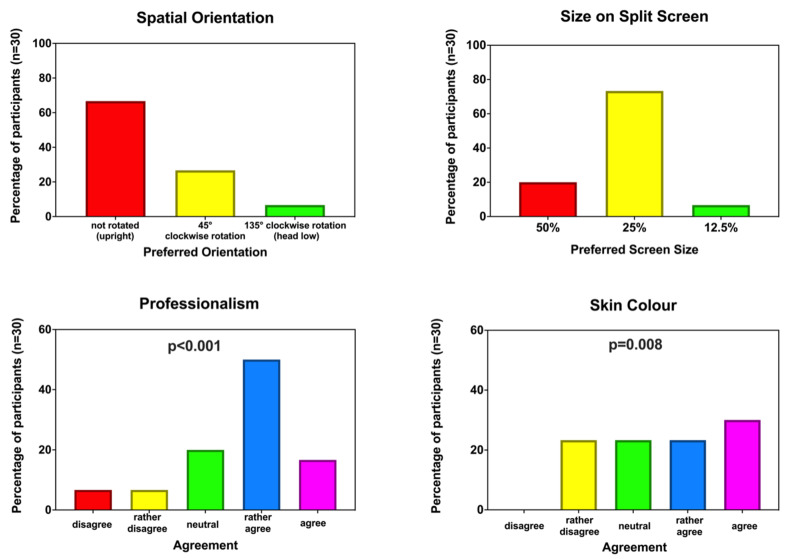
Responses to the general questions in the third, quantitative study part. The *p*-value indicates statistical difference from neutral (with Wilcoxon signed-rank test).

**Figure 5 diagnostics-12-00555-f005:**
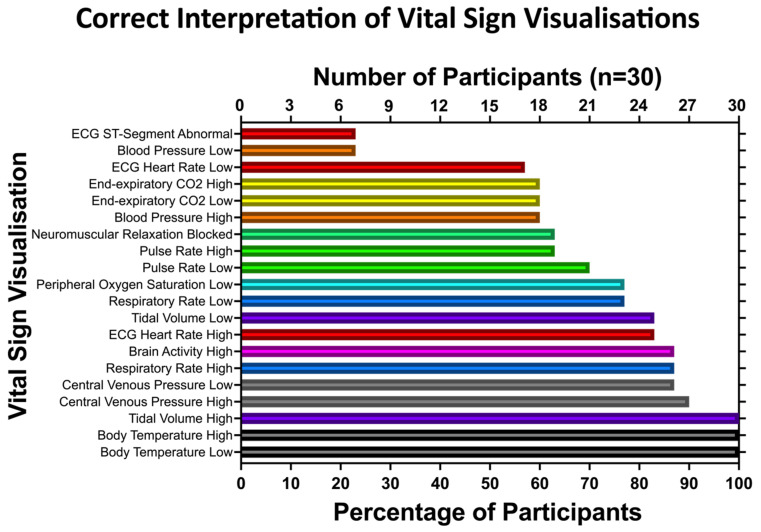
Intuitive interpretation accuracy per vital sign deviation. The correct interpretations are given as answer count and percentages. A maximum of 30 out of 30 (100%) interpretations per vital signs were correct.

**Table 1 diagnostics-12-00555-t001:** Study and participant characteristics.

	First, Qualitative Study Part	Third, Quantitative Study Part
Study Design	Interview-Based Assessment	Computer-Based Survey
Number of participants; *N1* and *N2,* respectively.	51	30
Gender of participants; *n* female of *N1* (%) and n female of *N2* (%), respectively	31 of 51 (61%)	18 of 30 (60%)
Number of physicians; n of *N1 (%)* and n of *N2 (%),* respectively.	28 of 51 (55%)	24 of 30 (80%)
Nurse anaesthetists; n of *N1 (%)* and n of *N2 (%),* respectively.	23 of 51 (45%)	6 of 30 (20%)
Anaesthesia experience in years; median (IQR [range])	5.5 (2.0–7.5 [1.0–36.0])	2.0 (1.0–3.0 [1.0–30.0])

**Table 2 diagnostics-12-00555-t002:** Main themes/subthemes with statement count and exemplary responses. N = 51 participants.

Main Theme	Subtheme	Examples
Appearance of Visual- Patient-avatar (33 statements made by 28 of 51 [55%] participants)	Illustration of Visual-Patient-avatar (22 statements made by 21 of 51 [43%] participants)	Participant #3: Looks a bit childish, very simple design. Participant #6: Still seems somewhat artificial. Participant #10: I feel like I’m in kindergarten at the moment when I see the figure. Participant #14: It has a very big head in contrast to the rest of the body. Participant #16: Is not very truthfully portrayed. Participant #21: A bit bare, dull.
	Spatial orientation of Visual-Patient-avatar (11 statements made by 11 of 51 [22%] participants)	Participant #17: It’s a bit unusual to see it like that. Um especially because it’s upside down. Participant #28: The position confuses me a bit. Participant #30: A patient who is lying upside down. Participant #32: I honestly have trouble with him lying with his head down.
General impression of Visual-Patient-avatar (26 statements made by 23 of 51 [45%] participants)	Positive comments (14 statements made by 14 of 51 [28%] participants)	Participant #4: Yes, looks fun. Participant #6: Really funny. Something positive. Participant #7: Quite funny, yes. Participant #14: Okay, it is cute. Participant #33: At first glance, probably rather a funny figure.
	Negative comments (12 statements made by 12 of 51 [24%] participants)	Participant #21: So, I find the picture quite unclear. I do not know what it wants to tell me. Participant #27: At the moment, I could still imagine having trouble with that, if that’s all I see. Participant #32: I am a bit irritated by the presentation. Participant #36: No idea. I do not know what to do with it.

## Data Availability

The original data are available in the Appendix A.

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
