# Peer review of "Improving Visual-Patient-Avatar Design Prior to Its Clinical Release: A Mixed Qualitative and Quantitative Study"

_diagnostics, 2022, doi:10.3390/diagnostics12020555_

Round 1

Reviewer 1 Report

The manuscript presents a Visual-Patient-avatar system that aims to promote situation awareness through user-centred design. 
The results of vital sign visualisations showed deficits in several vital sign visualizations, in particular central venous pressure.
I find the topic interesting and being worth of investigation and the document is well strucutred, organized, fluidly written, has enough background information, the methodology followed is clearly explained, the results are clearly presented and support the conclusions.
Although I propose the following suggestions:
- I strongly suggest authors from refraining using personal pronouns such as "we" and "our" throughout the text and I encourage them to write it in an impersonal form of writing.
- Abstract requires structuring such as: problem, motivation, aim, methodology, main results, further impact of those results.
- keywords should be in alphabetical order.
- Limitations should be added to the discussion section.
- A conclusion section should be added with the main conclusions and further proposed research 
- The link at appendiz 3 is missing.

Author Response

Reviewer 1:

Comment: The manuscript presents a Visual-Patient-avatar system that aims to promote situation awareness through user-centred design. The results of vital sign visualisations showed deficits in several vital sign visualizations, in particular central venous pressure.
I find the topic interesting and being worth of investigation and the document is well strucutred, organized, fluidly written, has enough background information, the methodology followed is clearly explained, the results are clearly presented and support the conclusions.

Response: Thank you so much for the favorable review of our technology and paper. We hope that we successfully incorporated your suggestions and improved our manuscript. Please find the point-by-point responses to the comments hereafter.

Comment: Although I propose the following suggestions:
- I strongly suggest authors from refraining using personal pronouns such as "we" and "our" throughout the text and I encourage them to write it in an impersonal form of writing.

Response: Thank you for this suggestion. We have corrected the manuscript accordingly and consistently refrain from using personal pronouns in the revised manuscript.

Comment: - Abstract requires structuring such as: problem, motivation, aim, methodology, main results, further impact of those results.

Response: Thank you very much. We have now tried to strengthen the abstract according to your suggestions and hope we have succeeded. In general, we have tried to strengthen the global overview and context in the abstract.

Comment:
- keywords should be in alphabetical order.

Response: Thank you. We have changed this as suggested.

Comment:

- Limitations should be added to the discussion section.

Response: Thank you very much for this suggestion. We have adjusted this according to your comment in the revised manuscript.

Comment:

- A conclusion section should be added with the main conclusions and further proposed research 

Response: Thank you very much. Have added a main conclusion section and strengthened the further proposed research section in the revised manuscript.

Comment:

- The link at appendiz 3 is missing.

Response:

Thank you very much for this comment. We added the link to the revised manuscript, and I am talking to the journal about hosting the video on its website to ensure its long-term availability.

Reviewer 2 Report

The authors present a timely and interesting study, but some modifications are required before publication.

My remarks are the following:

The English used is good, it requires only a minor modification. However, the dot should be removed from the end of the title.

The abstract should be rewritten to have a better flow of text. The "Background:", "Methods:", "Results": and "Conclusions:" words should be omitted from it.

The number of references should be increased, and the "Visual-Patient avatar" should be defined in more detail. How is it used exactly? What is its application? Why is it good that this avatar was modified? The answers to these questions should be included in the article.

Regarding the Wilcoxon tests, did the authors check for normality beforehand? Was the nonparametric distribution of data the reason to choose this statistical test?

Regarding the Likert scales, the authors need to check their Cronbach Alphas to know whether these Likert scales are valid enough.

The format of citations should be changed. They should not be written as superscript numbers, but as [1], [2,3], or [4-6] in the text. Also, the format of references should also be modified: journal names and volumenes should be italic, the years in journals bold, there should be a dot after surnames, and semicolons among authors. If possible, doi links can be included as well.

The link for the first video is not available in the Appendix.

Author Response

Reviewer 2:

Comment: The authors present a timely and interesting study, but some modifications are required before publication.

Response: Thank you for your work with our manuscript and the favorable feedback.

My remarks are the following:

The English used is good, it requires only a minor modification. However, the dot should be removed from the end of the title.

Response: We removed the dot at the end of the title as requested.  

Comment: The abstract should be rewritten to have a better flow of text. The "Background:", "Methods:", "Results": and "Conclusions:" words should be omitted from it.

Response: Thank you so much for this comment. We have now tried to strengthen the abstract according to the feedback of this peer review and hope we succeeded. We also removed the section words “Methods”… according to your suggestion.

Comment: The number of references should be increased, and the "Visual-Patient avatar" should be defined in more detail. How is it used exactly? What is its application? Why is it good that this avatar was modified? The answers to these questions should be included in the article.

Response: Thank for this very valuable input. We have now answered the questions in a new section in the introduction (Visual-Patient-avatar) and increased the number of references.

Comment: Regarding the Wilcoxon tests, did the authors check for normality beforehand? Was the nonparametric distribution of data the reason to choose this statistical test?

Response: Thank you very much. Yes, we examined the data for normal distribution and decided to use the one-sample Wilcoxon test because the data were not normally distributed.

Comment: Regarding the Likert scales, the authors need to check their Cronbach Alphas to know whether these Likert scales are valid enough.

Response: Thank you very much for this comment. Keeping the possibility of a misunderstanding in mind, we do not think that Cronbach's alpha would be appropriate in this case. We did not assess one construct with the two Likert-scale questions but two different constructs. Our Likert scales addressed whether the participants perceived skin colour to be representative of a global population and whether the design was perceived to be professional. The other two questions were not Likert scales. Please advise if you still want us to perform the analysis.

Comment: The format of citations should be changed. They should not be written as superscript numbers, but as [1], [2,3], or [4-6] in the text. Also, the format of references should also be modified: journal names and volumenes should be italic, the years in journals bold, there should be a dot after surnames, and semicolons among authors. If possible, doi links can be included as well.

Response: Thank you very much. Adjusted and added as suggested.

Comment: The link for the first video is not available in the Appendix.

Response: Thank you very much for this comment. We re-added the link to the revised manuscript, and I am talking to the journal about hosting the video on its website to ensure its long-term availability.

Reviewer 3 Report

The paper is well-written and has great future scope in improving the visual designs. The following remakes are made for this manuscript for further improvements:

  1. The use of pronoun (i.e. we) is not appropriate in the abstract.
  2. Illustration of methodology through the flow diagram is recommended.
  3. A data sample should also be placed in the annexure for reader’s reference.
  4. What are the criteria adopted while designing the qualitative survey? Mention it.

Author Response

Reviewer 3:

The paper is well-written and has great future scope in improving the visual designs. The following remakes are made for this manuscript for further improvements:

Comment:

  1. The use of pronoun (i.e. we) is not appropriate in the abstract.

Response: Thank you so much. We now refrain from using pronouns in the manuscript.

Comment:

  1. Illustration of methodology through the flow diagram is recommended.

Response: Thank you very much. We created and provided a flow chart now.

Comment:

  1. A data sample should also be placed in the annexure for reader’s reference.

Response: Thank you for pointing this out. We now provide the complete original interview transcripts. Also, we now provide the original data set for the results of the quantitative part.

Comment:       

  1. What are the criteria adopted while designing the qualitative survey? Mention it.

Response: Thank you for this comment. We added a paragraph in the Methods section (Description of qualitative part) where we discuss  why we decided to conduct the qualitative interviews in the chosen form.

Round 2

Reviewer 2 Report

The article has improved significantly.